# Efficacy of *Lactobacillus animalis* and *Propionibacterium freudenreichii*-Based Feed Additives in Reducing *Salmonella*-Associated Health and Performance Effects in Commercial Beef Calves

**DOI:** 10.3390/antibiotics11101328

**Published:** 2022-09-29

**Authors:** Charley Cull, Vijay K. Singu, Brooke J. Cull, Kelly F. Lechtenberg, Raghavendra G. Amachawadi, Jennifer S. Schutz, Keith A. Bryan

**Affiliations:** 1Midwest Veterinary Services, Inc., Oakland, NE 68045, USA; 2Central States Research Centre, Inc., Oakland, NE 68045, USA; 3Department of Clinical Sciences, College of Veterinary Medicine, Kansas State University, Manhattan, KS 66506, USA; 4Chr. Hansen, Inc., Milwaukee, WI 53214, USA

**Keywords:** cattle, food-safety, health, performance, probiotics, *Salmonella*

## Abstract

*Salmonella enterica*, which causes typhoid fever, is one of the most prevalent food-borne pathogens. Salmonellosis in cattle can greatly impact a producer’s income due to treatment costs, decreased productivity of the herd, and mortality due to disease. Current methods of treatment and prevention for salmonellosis consist of antibiotics and vaccinations, but neither of these options are perfect. Probiotics, categorized as antibiotic alternatives, are living microorganisms that are added to animal feeds in appropriate quantities in order to benefit health and productivity in adult and newborn livestock. The objective of this study was to demonstrate that *Lactobacillus animalis* and *Propionibacterium freudenreichii*, when used as a direct-fed microbial, was effective in reducing the adverse effects of experimentally induced *Salmonella* infection in beef calves. We conducted a single site efficacy study with masking using a randomized design comprising two groups of ten beef calves allocated to two treatment groups (control and probiotic). Procedures such as determining general health scores and body weight and collecting fecal samples were carried out following the experimental challenge of calves with *Salmonella* Typhimurium. The presence of at least one CFU of bacteria in feces was significantly higher among animals in the control than in the probiotic group, which was higher on days 0 to 7 than on days 8 to 14 (*p* = 0.012). Animals in the control group had a significantly higher presence of abnormal diarrhea scores than animals in the probiotic group (*p* < 0.001). Most notably, other health benefits in probiotic-fed group calves were obviously better than those for control calves and further substantiates the potential economic and health benefits of feeding effective probiotics.

## 1. Introduction

*Salmonella enterica*, which causes typhoid fever, is one of the most prevalent food-borne pathogens, and Salmonellosis causes 150,000 deaths each year [1,2]. Looking past human health, *Salmonella* also plays a significant role in bovine health. Salmonellosis in cattle can greatly impact a producer’s income due to treatment costs, decreased productivity of the herd, and mortality due to disease [3]. While there are 2500 serotypes, the most common is Typhi, which causes Salmonellosis in humans [4]. Another common serotype is Typhimurium, which causes Salmonellosis in mice, and *Salmonella* is often found in animal bacterial infections [4]. Current methods of treatment for salmonellosis consist of antibiotics and vaccinations, but neither of these options are perfect [1]. While vaccination is subpar, antibiotics could lead to unintentional selection for antibiotic resistant *Salmonella*, in addition to potentially hindering intestinal microflora [5]. However, antibiotic feed additives have become increasingly popular due to the desire to promote growth while also preventing infections [6]. These feed additives have contributed significantly to the emergence of antibiotic resistant bacteria [7].

Probiotics, categorized as antibiotic alternatives, are living microorganisms that are added to animal feeds in appropriate quantities to benefit health and productivity in adult and newborn livestock [8,9,10]. The microbial species most used as probiotics are lactic acid bacteria (LAB), most notably the *Lactobacillus* spp. and *Bifidobacterium* spp. strains. LAB have been used in *Salmonella* inhibition studies, and one of these studies have focused on bacteriocins as a treatment for food poisoning [11]. These studies have also focused on using coaggregation, auto aggregation, and intestinal cell and bacterial adhesion to hydrocarbons tests to decrease pathogens in the colon, which prevents food poisoning [12]. In vitro procedures using growth medium and tissue culture have also been used to examine the use of probiotics in the treatment of *Salmonella* [13,14]. Probiotics have been shown to be beneficial in small animal gastrointestinal infections, particularly *Escherichia coli* O157:H7 infections in rabbits and kittens [15], and *Helicobacter pylori* infections in small animals with gastric inflammation [16]. The effects of probiotics against *Salmonella* spp. infections have varied, but *Lactobacillus salivarius*, a LAB, has shown promise with *S. enterica* serovar Enteritidis in chickens where complete exclusion was noted by day 21 of treatment [17]. It has also been noted that a commercial probiotic mixture significantly decreased the mortality rate of chicks infected with S. enterica serovar Gallinarum [18].

The objective of this study was to demonstrate that *Lactobacillus animalis* and *Propionibacterium freudenreichii*, when used as a direct-fed microbial, was effective in reducing the adverse effects of experimentally induced Salmonella infection in beef calves.

## 2. Results

### 2.1. Performance Outcomes

#### 2.1.1. Body Weight Gain

The effect of treatment on body weight gain did not depend on study day, as depicted by a non-significant interaction between treatment and study day (*p* = 0.975). When modelling main effects only, treatment was not significantly (*p* = 0.923) associated with body weight gain. Study day, however, was significantly associated with the outcome (*p* < 0.001); specifically, body weight gain was much higher on day 43 than on day 0 (Table 1).

#### 2.1.2. Average Daily Gain (ADG)

Treatment was not significantly (*p* = 0.975) associated with ADG (Table 1).

### 2.2. Diagnostic Outcomes

Descriptive statistics for diagnostics (concentration of bacteria in feces) by treatment group and study day are depicted in Table 2.

#### Presence of at Least One CFU of Bacteria in Feces

The effect of treatment group on the presence of at least one CFU of bacteria in feces did not significantly vary by study day (interaction *p*-value = 0.999). When modeling the main effects only, the effect of treatment (*p* < 0.001) and study day (*p* = 0.016) were significantly associated with the presence of at least one CFU of bacteria in feces (modeled as a dichotomous response: Yes (>1 CFU/g), No (0 CFU/g)). The presence of at least one CFU of bacteria in feces was significantly higher among animals in control than in probiotic group. Moreover, the presence of at least one CFU of bacteria in feces was significantly higher on days 0 to 7 than on days 8 to 14 (*p* = 0.012) (Table 2 and Table 3: see significant contrasts). Similar results were obtained when considering results from days 0 to 14 only (Table 2 and Table 3).

### 2.3. Clinical Outcomes

#### 2.3.1. General Impression (Dichotomous)

The effect of treatment on the presence of abnormal general impression scores did not significantly depend on study day (interaction *p*-value = 0.169). Considering main effects only, the effect of treatment was significantly associated (*p* < 0.001) with the presence of abnormal general impression scores: animals in the control group had a significantly higher presence of abnormal general impression scores than animals in the probiotic group (Table 4). The presence of abnormal general impression scores significantly varied by study day (*p* < 0.001): presence of abnormal impression scores was higher on days 8 to 14 than on days 15 to 28 (*p* < 0.001), and higher on days 0 to 7 than on days 15 to 28 (*p* < 0.001) (Table 4). When considering the results from days 0 to 14 only, the treatment group was significantly associated with the presence of abnormal general impression scores (*p* < 0.001), but the study day was not (*p* = 0.097) (Table 4).

#### 2.3.2. Appearance (Dichotomous)

The effect of treatment on the presence of abnormal appearance scores did not significantly depend on study day (interaction *p*-value = 0.597). Considering main effects only, the effect of treatment was significantly associated (*p*< 0.001) with the presence of abnormal appearance scores: animals in the control had a significantly higher presence of abnormal appearance scores than animals in the probiotic group (*p*< 0.001) (Table 4). The presence of abnormal appearance scores did not significantly vary by study day (*p* = 0.232) (Table 4). Similar results were obtained when considering results from days 0 to 14 only (Table 4).

#### 2.3.3. Skin Tent (Dichotomous)

The effect of treatment on the presence of abnormal skin tent scores did not significantly depend on study day (interaction *p*-value = 0.634) when considering the results over the extended study period (days 0 to 28). Considering main effects only, both treatment group and study day were significantly associated (*p* < 0.001) with the presence of abnormal skin tent scores: the presence of abnormal skin tent scores was significantly higher among animals in the control than in the probiotic group. There was a higher percentage of abnormal skin tent scores on days 8 to 14 compared to days 0 to 7 (*p* < 0.001) and days 15 to 28 (*p* < 0.001) (Table 4: see significant contrasts).

When considering the results from days 0 to 14 only, the effect of treatment on the presence of abnormal skin tent scores significantly depended on the study day (interaction *p*-value = 0.055). The presence of abnormal skin tent scores was significantly higher among animals in control on days 8 to 14 than on days 0 to 7 (*p*< 0.001). Moreover, the presence of abnormal skin tent scores was significantly higher among animals in the control on days 8 to 14 than among animals in the probiotic group on the same study days (*p*< 0.001). Lastly, the presence of abnormal skin tent scores was significantly higher among animals in control on days 8 to 14 than among animals in probiotic on days 0 to 7 (Table 4: See Significant contrasts).

#### 2.3.4. Dehydration (Dichotomous)

The effect of treatment on the presence of abnormal dehydration scores did not significantly depend on study day (interaction *p*-value = 0.197) when considering the results over the extended study period (days 0 to 28). Considering main effects only, both treatment group and study day were significantly associated (*p* < 0.001) with the presence of abnormal dehydration scores. The presence of abnormal dehydration scores was significantly higher among animals in the control than in the probiotic group (*p* < 0.001), and higher on days 8 to 14 than on days 0 to 7 (*p* < 0.001) and days 15 to 28 (*p* < 0.001) (Table 5: see significant contrasts). When considering the results from days 0 to 14 only, the effect of treatment on the presence of abnormal dehydration scores significantly depended on the study day (interaction *p*-value = 0.055): the presence of abnormal dehydration scores was significantly higher among animals in control group on days 8 to 14 than on days 0 to 7 (*p* < 0.001). Moreover, the presence of abnormal dehydration scores was significantly higher among animals in the control on days 8 to 14 than among animals in the probiotic group on the same study days (*p* < 0.001). Lastly, the presence of abnormal dehydration scores was significantly higher among animals in the control on days 8 to 14 than among animals in the probiotic group on days 0 to 7 (Table 5: See Significant contrasts).

## 3. Discussion

*Salmonella* is a very important foodborne pathogen that affects both human and animal health. The study described here is unique as it generated data on efficacy of feeding two probiotic bacteria together on the clinical, health performance, and diagnostic outcomes among beef calves. The clinical challenge studies in calves associated with *Salmonella* and probiotic bacteria are very sparse.

An earlier study investigated the effect of probiotics on food intake and bacteria population in the feces. Calves were given probiotics daily, and there was a significant difference (*p* < 0.05) in body weight gain and feed efficiency [19]. Another study found that body weight gain was not significant for the first and second months between the treatment and control group, but it was significant between these groups during the third month. Based on the results from this study, probiotics benefited the calves used in the experiment, especially at three months of age. One-week old calves fed a milk replacer containing a probiotic with six strains (*Lactobacillus acidophilus*, *Lactobacillus salivarius*, *Lactobacillus paracasei*, *Lactobacillus plantarum*, *Lactococcus lactis*, and *Enterococcus faecium)* reported increased BW gain during the first 2 weeks of use, reduced incidence of diarrhea, and lower mortality [20]. Supplementing finishing pigs with *Bacillus subtilis* and *Clostridium butyricum* dietary probiotics increased growth performance throughout the entire experiment [21]. The ADG was larger when calves were given probiotics, prebiotics, and synbiotics at 6, 7, and 8 weeks (*p* < 0.05) [22].

A study on the herd prevalence of fecal *Salmonella* shedding and *Salmonella* Cerro accounted for 56% of the isolates, and 77% of the herds were positive for *Salmonella.* Authors are opined that cows can be asymptomatic carriers that can shed bacteria, which makes it challenging to control transmission of this serovar [23]. The DFM significantly reduced the probability of new infections with *Salmonella* among DFM-treated cattle compared with the controls (nontreated cattle) [24]. A study on the supplementation of *Lactobacillus animalis* and *Propionibacterium freudenreichii* as a direct-fed microbial (DFM) in feedlot cattle showed reductions in the prevalence and concentration of *Salmonella* in peripheral lymph nodes (PLNs) [25]. In another study, a 1:2 ratio of *Lactobacillus reuteri* and other *Lactobacillus* strains indicated little to no effect of DFMs on *Salmonella* in cattle, but an increase in the duration of administration to that similar to the duration used for commercial cattle might show treatment differences [26]. No treatment differences (*p* > 0.05) were observed for *Salmonella* in the PLNs except for the inguinal nodes, which had a significantly lower *Salmonella* prevalence in DFM-supplemented cattle than in the controls. Immune function was decreased (*p* < 0.05) in the treatment groups, which was measured by monocyte nitric oxide production and neutrophil oxidative burst [26]. This research indicated little to no effect of DFMs on *Salmonella* in cattle, but an increase in the duration of administration to that similar to the amount used for commercial cattle might show treatment differences [26]. Calves challenged with *Salmonella* Typhimurium and supplemented with a probiotic blend of *Lactobacillus casei* and *Enterococcus faecium* had reduced systemic production of haptoglobin and serum urea nitrogen but improved histomorphology of the duodenum and ileum [27].

Another study showed that *Saccharomyces cerevisiae* fermentation prototype inhibits the shedding, lymph node carriage, downstream virulence, and antibiotic resistance of *Salmonella* residing in cattle beyond the standard conventional practice, which includes monensin, tylosin, and a direct-fed microbial [28]. It was also noted that newborn piglets infected with *S. enterica* serovar Choleraesuis shed less pathogens and that there was reduced colonization in the lower intestine when treated with a similar culture [29]. However, the effects of these reduced pathogen numbers on the symptoms of the infection were not reported. Serovar Choleraesuis-infected piglets had decreased cecal and ileocolic junction *Salmonella* numbers when treated with a competitive exclusion culture of swine origin. It was also noted that *Salmonella*-positive tissues within the gut were decreased as well, but none of the animals exhibited signs of clinical symptoms of the infection [30]. In contrast, *E. coli* O157:H7 in lambs was not reduced by the probiotic *Lactobacillus acidophilus* [31], but a combination of *L. acidophilus* and *Streptococcus faecium* or just *Streptococcus* decreased the numbers of the pathogen. This was achieved by a probiotic mixture of *L. acidophilus*, *S. faecium*, *L. casei*, *Lactobacillus fermentum*, and *Lactobacillus plantarum*. Newborn piglets had decreased mortality and decreased enterotoxigenic *E. coli* shedding with a competitive exclusion culture [32]. A study on the effects of probiotics on *Salmonella* colonization in the ceca and various internal organs in laying hens challenged with *Salmonella enterica* serovar reported that probiotics regulate essential immune-cytokines, which aids in *Salmonella* control [33]. A study conducted in pullets discovered a reduction in *Salmonella* in the ceca of the *Bacillus* spp. probiotic treatment group, which demonstrates a possible pre-harvest food safety intervention by reducing a load of *Salmonella* in the ceca [34]. In contrast, a pig study reported that probiotic feed additive *C. butyricum* did not significantly reduce fecal excretion, serological response, intestinal carriage, or prevalence of *S.* Typhimurium [35]. *Salmonella enterica*-challenged pigs with a probiotic mixture containing *Lactobacillus murinus*, *Lactobacillus pentosus, Lactobacillus salivarius,* and *Pediococcus pentosaceus* had a correlation with reduced numbers of fecal *Salmonella* [36]. In contrast, calves supplemented with probiotics had a lower prevalence of diarrhea and lower fecal scores compared to the control group (*p* < 0.05) [37]. Another study with calves fed a garlic–probiotic mixture to reduce diarrhea reported higher final body weight, lower fecal scores, and fewer days of diarrhea compared to the control group [38]. However, adding a probiotic strain of *E. faecium* to the diet of weaned piglets challenged with *Salmonella* Typhimurium did not result in significant differences in clinical symptoms compared to animals not treated with probiotic. Instead, they observed increases in incidence of liquid feces and elevated body temperature in the probiotic group [39].

## 4. Materials and Methods

All activities related to this study were reviewed and approved by the Institutional Animal Care and Use Committee of Midwest Veterinary Services, Inc. prior to study initiation (IACUC number MVS18044B).

### 4.1. Animals

Twenty (20) healthy beef calves were selected for inclusion in the study. Cattle were commercially sourced from a feedlot in Nebraska. Cattle that were in overall good health with no complicating disease reported were enrolled for this study. Cattle were free of any complicating diseases (i.e., negative for *Salmonella*) and deemed acceptable by a site veterinarian on the physical examination. All cattle enrolled in the study had access to veterinary care throughout the study, and all veterinary care was at the discretion of the Site Veterinarian or Investigator in consultation with the Study Monitor when possible. There were no abnormalities noted as no cattle required euthanasia or a necropsy diagnosis throughout the study. Calves were individually housed indoors on concrete floors with no nose-to-nose contact. The housing conditions were as per the “Guide for the Care and Use of Agricultural Cattle in Research and Teaching by the Federation of Cattle Science Societies.” The temperature and humidity of the facility were monitored daily.

### 4.2. Study Design and Testing of a Probiotic Product

The study consisted of two groups of ten calves that were allocated randomly between two treatment groups: control and probiotic. The individual cattle were considered the experimental unit. The test probiotic product belongs to Chr. Hansen. The probiotic product (formulation containing *Lactobacillus animalis* and *Propionibacterium freudenreichii*) was administered as a top dressing over the pre-meal feed (approximate dose was 1 × 10^9^ CFU per head per day in 2 g of lactose). On study days −13–−8 and on study days −7–28, each calf received 2 scoops of the probiotic product with an approximate dose of 2 × 10^9^ CFU per head per day in 2 g of lactose. Feed mixes were made up daily. Approximately, 1 cup (i.e., a scoop) of pre-meal feed and the scoop(s) of the product were placed in a Ziploc bag and mixed around to ensure that all the feed was coated. The Ziploc bag had an individual animal ID on the bag, and they were separated by treatment group since they were stored frozen until administered. The cattle received a non-medicated complete pellet ration typical of industry standards that met or exceeded the minimum daily nutrient requirement for cattle of this age and class (i.e., NRC (National Research Council), 8th Edition, and Nutrient Requirements of Beef Cattle). The cattle were fed once daily, sufficient to meet appetite, and received water ad libitum. Study personnel involved in the collection, recording, or interpretation of any data were masked to the treatment assignment of cattle.

### 4.3. Experimental Challenge of Calves with Salmonella Typhimurium

The *Salmonella* Typhimurium challenge material was prepared at the Central State Research Center (CSRC), Diagnostic Laboratory (Oakland, NE, USA). The isolate used in the present study was obtained from a recent clinical case of bovine enteric salmonellosis from a feedlot calf. The concentration of bacteria present in the challenge material was determined to be 1.45 × 10^9^ CFU per mL pre-challenge and 1.30 × 10^9^ CFU per mL post-challenge. All cattle (*n* = 20) were challenged orally with approximately 1.38 × 10^9^ CFU/mL of *Salmonella* using a 60 mL catheter tipped syringe (without a needle).

### 4.4. Health Monitoring

At the study test facility, cattle (*n* = 20) underwent a physical examination by a veterinarian and were deemed physically eligible for continuation on the study. The systematic evaluation included general behavior and appearance, central nervous system, integumentary system, musculoskeletal system, feet, cardiovascular and respiratory systems, mucus membranes, lymph nodes, gastrointestinal and genitourinary systems, and temperament. Body weights, health scores, and diarrhea scorings were recorded accordingly. All of the health assessments were performed by the same person under the supervision of a veterinarian.

### 4.5. Fecal Sample Collection

Fecal samples were collected directly from the rectum using a new glove for each calf on days −7, 0, 3, 5, 7, 11, 14, and 21. All samples were placed in a sterile Whirl-Pak^®^ bag (Fisher Scientific, Hampton, NH, USA), and labeled with the calf identification, study number, and date of collection. Fecal samples were transferred to the laboratory on ice, and all the fecal samples were tested for *Salmonella* using microbial plating methods.

### 4.6. Fecal Salmonella Count and Concentration

Approximately, 1 g of fecal sample was mixed with 9 mL of tetrathionate broth vortexed before performing 10-fold serial dilution. One hundred microliters of 10^−1^, 10^−2^, 10^−3^, 10^−4^, 10^−5^, and 10^−6^ dilution were plated in duplicate onto the Xylose-Lysine-Tergitol-4 (XLT) agar plates (Hardy Diagnostics, Santa Maria, CA, USA). All plates and enrichment tubes containing tetrathionate broth were incubated at 37 °C for approximately 24 h. All agar plates were evaluated the next day for viable cell counts. Samples which were negative for *Salmonella* were further evaluated by plating the 24 h enrichment cultures onto XLT agar and then incubated at 37 °C for approximately 24 h.

### 4.7. Statistical Analysis

The primary outcome variables were reductions in the severity and/or duration of clinical signs of disease associated with *Salmonella* infection that included but may not be limited to diarrhea, depression, dehydration, *Salmonella* concentration levels in feces, and reduced body weight. Health scores were secondary variables. Average daily weight gain was calculated as [(final body weight − initial body weight)/number of days between initial and final weight measurements] and considered all study subjects (*n* = 20). Descriptive statistics (mean, median, standard deviation, and range) for continuous and frequency tables for discrete outcomes were computed by treatment group and study day. Generalized linear mixed models (GLMM) were fitted to estimate the effect of treatment over time on clinical, production, and diagnostic outcomes. Continuous outcomes, body weight gain, and average daily weight gain were modeled with a Gaussian distribution using identity link and maximum likelihood estimation. Dichotomous outcomes (yes/no), including the presence of at least one CFU of *Salmonella* spp. in feces, and clinical scores (diarrhea score, general impression, skin tent, dehydration, and appearance scores) were modeled using a binary distribution, logit link, and restricted pseudo-likelihood estimation, using PROC GLIMMIX in SAS 9.4 (SAS Institute Inc., Cary, NC, USA). A multivariable model including fixed effects for the treatment group, study day, and a two-way interaction term between treatment group and study day were fitted against performance, diagnostic, and clinical outcomes. When the interaction term was not significantly associated with the outcome (*p* > 0.05), a model with main effects only (treatment group and study day) was fitted. Models included a first-order autoregressive or a heterogeneous autoregressive covariance structure, when applicable, for animal id to account for repeated measures (equally or unequally spaced, respectively). For ADG, a univariable model evaluating the fixed effect of treatment was fitted. *p*-values < 0.05 were considered statistically significant. Means and mean percentages, standard error of the mean, 95% confidence intervals, and *p*-values were reported. The Tukey–Kramer adjustment for multiple comparisons was used to prevent inflation of the type I error. Model fit and distributional assumptions were evaluated using graphical and (statistical) test approaches.

## 5. Conclusions

*Salmonella enterica,* which causes typhoid fever, is one of the most prevalent food-borne pathogens. Salmonellosis in cattle can greatly impact a producer’s income due to treatment costs, decreased productivity of the herd, and mortality due to disease. Daily feeding of a probiotic product after an oral challenge with *Salmonella* Typhimurium significantly improved both the general impression and appearance scores of calves. Most notably, other health benefits in probiotic-fed group calves were significantly better than those for control calves and further substantiates the potential economic and health benefits of feeding effective probiotics.

## Figures and Tables

**Table 1 antibiotics-11-01328-t001:** Body weight gain and ADG of commercial beef calves supplemented with *Lactobacillus animalis* and *Propionibacterium freudenreichii*-based feed additives to reduce *Salmonella*.

	Body Weight Gain (kg)				Average Daily Gain (kg)			
Variable	Mean	SEM	95% CI	*p*-Value	Mean	SEM	95% CI	*p*-Value
**Treatment**				0.923				0.975
Probiotics	254.6	10.7	232.2–277.1		0.54	0.08	0.38–0.70	
Control	253.2	10.7	230.7–275.6		0.55	0.08	0.39–0.71	
**Study Day**				**<0.001**				
0	242.2	7.6	226.2–258.2					
43	265.6	7.6	249.6–281.6					

**Table 2 antibiotics-11-01328-t002:** Concentration of *Salmonella* in feces by treatment group and study day in commercial beef calves supplemented with *Lactobacillus animalis* and *Propionibacterium freudenreichii*-based feed additives to reduce *Salmonella*.

	Concentration of *Salmonella* in Feces (CFU/g)		Presence of at Least One CFU of Bacteria in Feces	
	*n*	Mean	Pos (≥1 CFU/g)	Neg (0 CFU/g)
**Treatment**				
Probiotics	70	92,227.9	17	53
Control	70	296,317.1	43	27
**Study Day**				
0	20	0.0	0	20
3	20	1,100,675.0	18	2
5	20	188,200.0	17	3
7	20	23,847.5	12	8
9	20	47,105.0	10	10
14	20	80.0	3	17
28	20	0.0	0	20

*n* = number of observations.

**Table 3 antibiotics-11-01328-t003:** Presence of at least one CFU of *Salmonella* in feces of commercial beef calves supplemented with *Lactobacillus animalis* and *Propionibacterium freudenreichii*-based feed additives to reduce *Salmonella*
^a^.

	Presence of at Least One CFU in Feces (d 0–14)				Presence of at Least One CFU in Feces (d 0–28)			
Variable	Mean %	SEM	95% CI	*p*-Value	Mean %	SEM	95% CI	*p*-Value
**Treatment**				**<0.001**				**<0.001**
Probiotics	22.0	5.9	12.2–36.2		0.2	25.4	0–100	
Control	66.8	6.9	51.6–79.1		1.2	91.0	0–100	
**Study Day**				**0.009**				**0.016**
d 0–7	58.8	6.5	45.3–71.1		61.1	6.2	48.5–72.4	
d 8–14	28.4	8.0	15.3–46.6		28.5	7.8	15.7–45.9	
d 15–28					0.0	0.0	0–100	

^a^ Multivariable model estimating the effect of treatment over time on the presence of at least one CFU of bacteria in feces included fixed effects for treatment group and study day and a covariance structure to account for repeated measures at the animal level.

**Table 4 antibiotics-11-01328-t004:** Clinical scores of commercial beef calves supplemented with *Lactobacillus animalis* and *Propionibacterium freudenreichii*-based feed additives to reduce *Salmonella*
^a^.

	Clinical Scores											
	Diarrhea Scores (≥1 vs. 0)				General Impression Scores (≥1 vs. 0)				Appearance Scores (≥1 vs. 0)			
Variable	Mean %	SEM	95% CI	*p*-Value	Mean %	SEM	95% CI	*p*-Value	Mean %	SEM	95% CI	*p*-Value
**Treatment** **(d 0–14)**				**0.054**				**<0.001**				**0.002**
Probiotics	15.4	6.3	6.3–32.8		3.5	2.3	0.9–12.4		6.3	3.4	2.0–17.6	
Control	37.9	8.6	22.6–55.9		36.1	6.3	24.8–49.2		35.8	6.8	23.6–50.1	
**Treatment** **(d 0–28)**				0.147				**<0.001**				**<0.001**
Probiotics	0.09	11.4	0–100		1.3	0.7	0.5–3.9		0.03	4.0	0–100	
Control	0.30	36.6	0–100		15.7	3.4	10.0–23.7		0.30	32.5	0–100	
**Study Day**				0.997				**<0.001**				0.232
d 0–7	31.2	3.8	24.2–39.1		9.1	2.9	4.7–16.7		14.8	3.0	9.8–21.7	
d 8–14	30.8	4.0	23.5–39.3		15.1	4.1	8.6–25.2		22.4	3.9	15.6–31.0	
d 15–28	0.0	0.0	0–100		0.7	0.5	0.2–2.6		0.0	0.0	0.0–100	

^a^ Multivariable model estimating the effect of treatment group over time on clinical outcomes (each outcome modeled separately) included fixed effects for treatment group and study day, and a covariance structure to account for repeated measures at the animal level.

**Table 5 antibiotics-11-01328-t005:** Dehydration and skin tent scores in commercial beef calves supplemented with *Lactobacillus animalis* and *Propionibacterium freudenreichii*-based feed additives to reduce *Salmonella*
^a^.

Variable	Clinical Scores							
	Dehydration Scores (≥1 vs. 0)				Skin Tent Scores (≥1 vs. 0)			
	Mean %	SEM	95% CI	*p*-Value	Mean %	SEM	95% CI	*p*-Value
**Treatment (d 0–14)**				**0.003**				**0.003**
Probiotics	10.3	3.9	4.5–21.5		10.3	3.9	4.5–21.5	
Control	43.8	8.8	26.3–63.0		43.8	8.8	26.3–63.0	
**Treatment (d 0–28)**				**0.001**				**<0.001**
Probiotics	3.3	1.4	1.4–7.7		4.3	1.2	2.5–7.2	
Control	21.6	6.0	11.3–37.3		22.9	3.3	17.1–29.9	
**Study Day**				**<0.001**				**<0.001**
d 0–7	12.2	3.5	6.7–21.3		14.3	2.9	9.6–20.9	
d 8–14	36.1	6.9	23.4–51.1		37.7	4.6	29.3–47.0	
d 15–28	1.1	0.6	0.4–2.9		1.5	0.6	0.6–3.4	
**Treatment × Study Day**				**0.055**				**0.055**
Probiotics × d 0–7	8.0	3.7	3.0–19.5		8.0	3.7	3.0–19.5	
Control × d 0–7	24.2	7.5	12.0–42.8		24.2	7.5	12.0–42.8	
Probiotics × d 8–14	13.1	5.4	5.4–28.4		13.1	5.4	5.4–28.4	
Control × d 8–14	65.6	9.1	45.0–81.6		65.6	9.1	45.0–81.6	

^a^ Multivariable model estimating the effect of treatment over time on clinical outcomes (each outcome modeled separately) included fixed effects for treatment group and study day (and a two-way interaction term between treatment and study day for models including days 0 to 14 only) and a covariance structure to account for repeated measures at the animal level.

## Data Availability

The data presented in this study are available from the corresponding author upon reasonable request.

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
