# Peer review of "Efficacy of Lactobacillus animalis and Propionibacterium freudenreichii-Based Feed Additives in Reducing Salmonella-Associated Health and Performance Effects in Commercial Beef Calves"

_antibiotics, 2022, doi:10.3390/antibiotics11101328_

Round 1

Reviewer 1 Report

Salmonellosis nowadays represents a major threat to bovine health. Salmonella enterica, which causes typhoid fever, is one of the most prevalent food-borne pathogens, and Salmonellosis causes 150,000 deaths annually. This is a nice paper regarding the dietary supplementation of Lactobacillus animalis and Propionibacterium freudenreichii, probiotics when used as a direct-fed microbial, as an antibiotic alternative to reduce the adverse effects of experimentally induced Salmonella infection in beef calves.

The study described here is unique as it generated data on the efficacy of feeding two probiotic bacteria together on the clinical, health performance, and diagnostic outcomes among beef calves. The clinical challenge studies in calves associated with Salmonella and probiotic bacteria are very sparse.

The introduction is well written as is the materials and methods section. The results section is more analytic and descriptive of the tables. The discussion is solid and well referenced. 

Line 21 in the abstract section: the names of Lactobacillus animalis and Propionibacterium freudenreichii should be italic.

Overall, nice work. Congratulations

Author Response

Salmonellosis nowadays represents a major threat to bovine health. Salmonella enterica, which causes typhoid fever, is one of the most prevalent food-borne pathogens, and Salmonellosis causes 150,000 deaths annually. This is a nice paper regarding the dietary supplementation of Lactobacillus animalis and Propionibacterium freudenreichii, probiotics when used as a direct-fed microbial, as an antibiotic alternative to reduce the adverse effects of experimentally induced Salmonella infection in beef calves.

The study described here is unique as it generated data on the efficacy of feeding two probiotic bacteria together on the clinical, health performance, and diagnostic outcomes among beef calves. The clinical challenge studies in calves associated with Salmonella and probiotic bacteria are very sparse.

The introduction is well written as is the materials and methods section. The results section is more analytic and descriptive of the tables. The discussion is solid and well referenced. 

Thank you!

Line 21 in the abstract section: the names of Lactobacillus animalis and Propionibacterium freudenreichii should be italic.

Agreed and Complied.

Overall, nice work. Congratulations.

Thank you!

Reviewer 2 Report

At the outset, I would like to thank you for the opportunity to participate in the review of the manuscript.

The introduction to the article is clearly written, introduces the topic well, the selection of literature is appropriate (although many items are too old).

The purpose and scope of the article have been correctly defined. The methodology is clearly written and makes it possible to repeat the experiments. The results and discussion are well presented.

I consider the entire manuscript interesting and worthy of attention. The manuscript was pleasant to read.

Below are some notes on the manuscript.

From 2020, a new nomenclature for lactic acid bacteria is in force. Please update the names throughout the manuscript. Check out the article: Zheng et al., Int. J. Syst. Evol. Microbiol. 2020; 70: 2782–2858 DOI 10.1099/ijsem.0.004107

Line 21, 22, 27, 38, 43, 56… - Please write the names of the bacteria in italics

Line 37-38 - "Salmonellosis causes 150,000 deaths each year (Das et al., 2013, Majowiez et al., 2010)" - This data is old, we are in 2022. Is this number lower or higher? Please update your literature, reports from WHO are a good source.

Line 61 - "in vitro" should be italicized

Line 91-92 - Abbreviations of species are used once and full names. Please standardize.

Line 106. Tables should be cited in the order in which they appear in the text. Why are you citing table 2 above table 1?

Table 1 - What is ADG? Abbreviations should be explained below the table.

Line 224 - Please expand the ADG abbreviation in the text.

Line 332-333 - Please specify the manufacturer of the microbiological medium.

Methodology - Please add the "Material" paragraph in which the applied probiotic strains will be described. Where were these strains isolated from? Were they genetically identified? Are their sequences in GenBank? Are they patentable? I understand that bacterial strains are part of Chr Hansen's trade secret? More information is necessary.

There are 57 items in the literature list, of which more than half (!) are items from 1987-2012. Please update literature. It is unacceptable to cite such a number of articles older than 10 years.

Author Response

Reviewer 2:

At the outset, I would like to thank you for the opportunity to participate in the review of the manuscript.

The introduction to the article is clearly written, introduces the topic well, the selection of literature is appropriate (although many items are too old).

Thank you! The literature selection is bit old since, not much has been done on this topic lately.  We think our study is very novel and unique.

The purpose and scope of the article have been correctly defined. The methodology is clearly written and makes it possible to repeat the experiments. The results and discussion are well presented.

Thank you!

I consider the entire manuscript interesting and worthy of attention. The manuscript was pleasant to read.

Thank you!

Below are some notes on the manuscript.

From 2020, a new nomenclature for lactic acid bacteria is in force. Please update the names throughout the manuscript. Check out the article: Zheng et al., Int. J. Syst. Evol. Microbiol. 2020; 70: 2782–2858 DOI 10.1099/ijsem.0.004107.

Agreed.  But, the present work is done on the commercially available product which is labeled and marketed as Lactobacillus animalis.  So, we would like to keep the same nomenclature if possible.  We don’t want to confuse the readers.

Line 21, 22, 27, 38, 43, 56… - Please write the names of the bacteria in italics.

Revised accordingly.

Line 37-38 - "Salmonellosis causes 150,000 deaths each year (Das et al., 2013, Majowiez et al., 2010)" - This data is old, we are in 2022. Is this number lower or higher? Please update your literature, reports from WHO are a good source.

Line 61 - "in vitro" should be italicized

Agreed and Complied.

Line 91-92 - Abbreviations of species are used once and full names. Please standardize.

Complied.

Line 106. Tables should be cited in the order in which they appear in the text. Why are you citing table 2 above table 1?

Thank you. Revised accordingly.

Table 1 - What is ADG? Abbreviations should be explained below the table.

Revised accordingly.

Line 224 - Please expand the ADG abbreviation in the text.

Revised accordingly.

Line 332-333 - Please specify the manufacturer of the microbiological medium.

Revised accordingly.

Methodology - Please add the "Material" paragraph in which the applied probiotic strains will be described. Where were these strains isolated from? Were they genetically identified? Are their sequences in GenBank? Are they patentable? I understand that bacterial strains are part of Chr. Hansen's trade secret? More information is necessary.

Added a sentence on the source of this product.

There are 57 items in the literature list, of which more than half (!) are items from 1987-2012. Please update literature. It is unacceptable to cite such a number of articles older than 10 years.

Thank you! The literature selection is bit old since, not much has been done on this topic lately.  We think our study is very novel and unique.  We struggled a bit to find relevant new references.

Reviewer 3 Report

Dear, after carefully considering the manuscript submitted for review my opinion is as follows.

This manuscript discussed effects of probiotics against Salmonella spp. when used as a direct-fed microbial infections in reducing adverse effects of experimentally induced Salmonella infection in beef calves. A prohibition of the use of antibiotics as growth promoters in livestock production systems is a worldwide initiative, due to arising antimicrobial resistance. Therefore, the manuscript is relevant to the field, work fits the journal scope, but in terms of novelty  manuscript does not provide a significant advancement of the current knowledge.

Regarding the structured manner, is not well organized and my opinion is inserted in the comments. The data is not interpreted appropriately and consistently throughout the manuscript, therefore must be improved. 

The tables are appropriate, they properly show the data, and they are easy to interpret and understand, but the highest standards for presentation of the results used are not achieved. All of the results are presented in the tables.

A similar review published recently by EFSA, Safety and efficacy of a feed additive consisting on Ligilactobacillus animalis ATCC PTA-6750 (formerly Lactobacillus animalis) for all animal species (Chr. Hansen A/S), EFSA Journal 2021;19(3):6469. According to the EFSA opinion The studies provided showed that L. animalis ATCC PTA-6750 when used in combination with Propionibacterium freudenreichii ssp. shermanii ATCC PTA-6752 has the potential to act as an acidity regulator. However, the Panel has reservations on the effects of this mixture in practical use conditions. In the absence of valid studies investigating relevant and specific endpoints, no conclusion can be drawn on the efficacy of L. animalis ATCC PTA-6750 as preservative or hygiene condition enhancer.

Considering EFSA conclusion, in terms of reservations on the effects of this mixture in normal practical use conditions, also could take reserve for obtained results. 

Conclusions are not justified and well supported by the results and thus must be improved. 

The majority of the cited references are older than 5 last years and some of them are not relevant.

The paper is interesting for the readership of the journal but must be improved according to the reviewer's suggestions

Author Response

Dear, after carefully considering the manuscript submitted for review my opinion is as follows.

This manuscript discussed effects of probiotics against Salmonella spp. when used as a direct-fed microbial infections in reducing adverse effects of experimentally induced Salmonella infection in beef calves. A prohibition of the use of antibiotics as growth promoters in livestock production systems is a worldwide initiative, due to arising antimicrobial resistance. Therefore, the manuscript is relevant to the field, work fits the journal scope, but in terms of novelty manuscript does not provide a significant advancement of the current knowledge.

Thank you!

Regarding the structured manner, is not well organized and my opinion is inserted in the comments. The data is not interpreted appropriately and consistently throughout the manuscript, therefore must be improved. 

We have revised and improved the manuscript in this version.

The tables are appropriate, they properly show the data, and they are easy to interpret and understand, but the highest standards for presentation of the results used are not achieved. All of the results are presented in the tables.

Thank you!

A similar review published recently by EFSA, Safety and efficacy of a feed additive consisting on Ligilactobacillus animalis ATCC PTA-6750 (formerly Lactobacillus animalis) for all animal species (Chr. Hansen A/S), EFSA Journal 2021;19(3):6469. According to the EFSA opinion The studies provided showed that L. animalis ATCC PTA-6750 when used in combination with Propionibacterium freudenreichii ssp. shermanii ATCC PTA-6752 has the potential to act as an acidity regulator. However, the Panel has reservations on the effects of this mixture in practical use conditions. In the absence of valid studies investigating relevant and specific endpoints, no conclusion can be drawn on the efficacy of L. animalis ATCC PTA-6750 as preservative or hygiene condition enhancer.

Considering EFSA conclusion, in terms of reservations on the effects of this mixture in normal practical use conditions, also could take reserve for obtained results. 

Thank you! Yes, we do agree that further studies are warranted to derive at the conclusion when using these products in animal systems.

Conclusions are not justified and well supported by the results and thus must be improved. 

We have revised the conclusion section accordingly.

The majority of the cited references are older than 5 last years and some of them are not relevant.

 Thank you! The literature selection is bit old since, not much has been done on this topic lately.  We think our study is very novel and unique.  We struggled a bit to find relevant new references.

The paper is interesting for the readership of the journal but must be improved according to the reviewer's suggestions.

Thank you!

Round 2

Reviewer 3 Report

Dear, following the first revision round, I think that the authors have significantly improved the manuscript according to the reviewer's suggestion, but some minor revisions are necessary before publishing. 

Author Response

Please, could you revise this part in order to be in a logical order and linked with the previous part? You were just copying and pasting.

We have revised and improved the manuscript in this version. Thank you!

As before, I could not conclude that statement from table 4.

Agreed and complied.

This is contradictory to EFSA Conclusions on efficacy. EFSA Journal 2021;19(3):6469

Could you check statements.

We have decided to delete this statement.  Thank you!